# Looks Great, Functions Better: Physics Compliance Text-to-3D Shape Generation

## Abstract

Text-to-3D shape generation has shown great promise in generating novel 3D content based on given text prompts. However, existing generative methods mostly focus on geometric or visual plausibility while ignoring functionality for the generated 3D shapes. This greatly hinders the practicality of generated 3D shapes in real-world applications. In this work, we propose Fun3D, a physics compliance functional text-to-3D shape generation method. By analyzing the solid mechanics of generated 3D shapes, we reveal that the 3D shapes generated by existing text-to-3D generation methods are impractical for real-world applications as the generated 3D shapes do not conform to the laws of physics. To this end, we leverage 3D diffusion models to provide 3D shape priors and design a data-driven differentiable physics layer to optimize 3D shape priors with solid mechanics. This allows us to optimize geometry efficiently and learn physics information about 3D shapes at the same time. Experimental results demonstrate that our method can consider both geometric plausibility and functional requirement, further bridging 3D virtual modeling and physical worlds.

## 1 Introduction

Text-to-3D shape generation can synthesize novel 3D content based on text prompts. It is critical for a variety of applications, including personalized customization, film-making, robotics simulation, gaming, and so on. Recently, text-to-3D shape generation Poole et al. (2022); Lin et al. (2023); Tang et al. (2023) have achieved promising results with the development of generative models (e.g., generative adversarial networks Goodfellow et al. (2020) , diffusion models Ho et al. (2020); Rombach et al. (2022)) and 3D representations (e.g., DeepSDF Park et al. (2019) and neural radiance fields Mildenhall et al. (2021)). Existing text-to-3D generation methods mostly focus on the visual realism of generated 3D shapes.

The remarkable performance of text-to-3D shape generation has not only led to significant advancements in the computer vision domain but has also generated substantial interest in real-world applications, such as the engineering design Guo et al. (2023); Rios et al. (2023). In contrast to computer vision, where the primary focus is on the visual aspects of 3D objects, real-world design problems demand additional considerations for physical functionality in real-world environments. For instance, in the design of mechanical components, it is crucial to incorporate features such as chamfered corners to mitigate stress concentration and avoid vertical transitions to enhance component longevity Sackfield et al. (2013). These requirements necessitate the integration of physical principles with visual aesthetics to ensure that the generated 3D objects are both practical and functional in real-world applications. However, most current methods fail to incorporate physical information, leading to 3D objects that are unfunctional for real-world environments. More seriously, due to physical deficiencies, the generative model may even generate ill-defined or unrealistic 3D-shapes, further obstructing the generative capability.

Real-world functionality requirements such as engineering designs commonly account for the interaction between 3D geometry and the physical environment, with physical information being key to describing this interaction. Given the importance of physics, this work emphasizes incorporating physical information into the generative process. Specifically, we treat the object described by the 3D geometry as a solid and incorporate basic solid mechanics properties into our analysis. Thus, we conform the 3D geometry to linear elasticity Bertram & Glüge (2015) behavior. In Figure 1,

Figure 1: **Motivation.** By applying distributed force on the object top, we analyze the solid mechanics of 3D shapes generated by text-to-3D generation methods by FEM. For physics, the lighter the color values, the higher the stress levels. We observe that the generated 3D shape by Diffusion-SDF Li et al. (2023) will experience notably higher stress in some regions, demonstrating that the generated geometry is fragile. In contrast, the stress distribution of our generated 3D shape is more uniform as the generated geometry conforms to the physical laws. Moreover, our learned physics information is comparable to the physics obtained by FEM.

we analyze the solid mechanics of a 3D shape generated by Diffusion-SDF using the conventional Finite Element Method (FEM) Zienkiewicz & Taylor (2000). By applying distributed force on the tabletop, we can observe that stress (quantified by the von-Mises stress) spreads across the entire geometry, with certain regions experiencing notably higher stress levels than others. Typically, these concentrated high-stress regions are potential areas of failure due to external forces. Therefore, minimizing the maximum stress and uniformly distributing the stress throughout the geometry can reduce susceptibility to failure, thus improving the functionality of the generated object. This paper aims to achieve geometry generation and optimization, considering not only computer vision but also physical functionality.

In this paper, we propose Fun3D, a physics compliance text-to-3D shape generation method to achieve the functional generation. Specifically, our method contains two stages. In the first stage, we initialize a 3D shape based on 3D diffusion models and convert it into implicit geometry representations, Signed Distance Function (SDF) Park et al. (2019). This allows our approach to be flexibly combined with existing generative methods. In the second stage, we design a data-driven differentiable physics layer to enable geometry and physics optimization at the same time. Our differentiable physics layer is parameterized by neural networks and used to learn the solid mechanics of generated 3D shapes on demand when optimizing geometries. To guarantee the precision of the feedback physics information, we introduce a physics prior embedding which is achieved by leveraging the FEM results computed for the initial generated 3D shape to initialize the differentiable physics layer. We introduce a relaxed geometry loss and a series of physics losses during training to guide the optimization. Moreover, we design an alternating training strategy to learn geometry and physics information collaboratively. In this way, our method can simultaneously optimize geometry and learn the physics of generated 3D shapes, considering both the geometric plausibility and the functionality requirement, i.e., *"What you get is what you text, and what you get is immediately functional"*.

Overall, our contributions are summarized as follows:

- We propose Fun3D, i.e., a physics compliance text-to-3D functional shape generation method, to incorporate the physical laws into generated 3D shapes. This approach bridges the gap between 3D virtual modeling and the physical world, ensuring that the generated shapes are both visually appealing and physically functional.

- We design a data-driven differentiable physical layer to learn physics cues during training effectively. Moreover, an alternating training strategy is introduced to stabilize the learning of physics. Thus, our method can collaboratively optimize geometry and physics flexibly and reliably.

- Experiments demonstrate that our method can improve geometric or visual preferences without requiring extra training data and manual interaction and endow generated 3D shapes with physical perception capabilities.

## 2 RELATED WORK

**Text-to-3D Shape Generation** Recently, text-to-3D shape generation has achieved significant success. On the one hand, some generation methods train their models on text-3D pairs and can directly

generate 3D shapes from texts without requiring any optimization at inference Sanghi et al. (2022); Mittal et al. (2022); Li et al. (2023); Nichol et al. (2022); Jun & Nichol (2023). CLIP-forge Sanghi et al. (2022) uses renderings of shapes with pre-trained image-text joint embedding to learn shapes conditioned on texts. AutoSDF Mittal et al. (2022) explores an auto-regressive prior for 3D shape generation based on a discretized SDF autoencoder. Recent works employ diffusion models to learn a probabilistic mapping from the text to 3D shapes, such as Diffusion-SDF Li et al. (2023), Point-E Nichol et al. (2022) and Shape-E Jun & Nichol (2023). On the other hand, some methods lift 2D diffusion models for text-to-3D generation Poole et al. (2022); Lin et al. (2023); Tang et al. (2023); Wang et al. (2024). These methods, such as DreamFusion Poole et al. (2022) and Dream-Gaussian Tang et al. (2023), leverage pretrained text-to-image diffusion models as the guidance to optimize 3D representations, including neural radiance fields (NeRF) Mildenhall et al. (2021); Barron et al. (2021); Müller et al. (2022); Fu et al. (2022) and Gaussian Splatting Tang et al. (2023). Therefore, these methods do not require any 3D data. However, whether these generation methods train their models on 3D data or text-to-image priors, there is no guarantee that they can perceive physics information from large amounts of data. This work shows that our method can learn physics properties for the generated 3D shapes during training.

**Physics-aware 3D Generation** There exist several physics-aware 3D generation methods Mezghanni et al. (2021; 2022); Liu et al. (2023b); Wang & He (2019); Xie et al. (2024); Zhong et al. (2024); Guo et al. (2024); Chen et al. (2024). These methods utilize either offline simulations or online simulations to ensure the physical validity of generated shapes. Mezghanni et al. (2021) pre-trains a surrogate network using offline simulations to predict stability to inform the generative learning. Mezghanni et al. (2022); Liu et al. (2023b) design online simulation layers to provide physical guidance for the generative learning. All these methods do not need to learn physics feedback during shape learning. Unlike them, our method collaboratively learns geometry and physics during training.

**Physics-Informed Machine Learning** The simulation and modeling of physical systems have long played a crucial role in various scientific and engineering domains Heermann & Heermann (1990). Traditionally, the FEM has been instrumental in these tasks. However, recent advancements in deep neural networks have prompted researchers to explore novel applications in these domains Raissi (2018). Among these methods, PINNs have garnered significant attention Raissi et al. (2019). PINNs stand out due to their ability to integrate governing equations of mechanics, often expressed through PDEs, into the learning objectives of neural networks. This integration ensures the network outputs align with the fundamental governing physics. Consequently, there has been a proliferation of studies on PINNs in recent years Deng et al. (2023); Chiu et al. (2022); Mao et al. (2020); Wang et al. (2023). While several methods have been proposed, the majority of research on PINNs primarily focuses on simulating physical systems. There is a limited number of discussions on optimization based on PINNs, such as physics-informed inverse design Lu et al. (2021), PDE constrained optimization Hao et al. (2022), and PINN-assisted topology optimization Jeong et al. (2023); Yin et al. (2023). However, most of these works concentrate on optimizing simple functions or 2D geometries. None of them address the optimization of 3D geometries.

## 3 METHOD

Given a text describing our target object, Fun3D aims to generate a 3D shape that simultaneously satisfies geometric plausibility and physical functionality. This section first gives an overview of the whole framework of Fun3D in Sec. 3.1. Then, we describe the geometry representations and initialization in Sec. 3.2. We elaborate on our physics representations and governing equations used to model the physical laws in Sec. 3.3. In Sec. 3.4, we detail our physics prior embedding and training process.

### 3.1 OVERVIEW

Figure 2 depicts the overall framework of our method. It is a two-stage pipeline, including initialization with diffusion-based methods and optimization with our designed differentiable physics layer. In the initialization stage, we generate an initial 3D shape by leveraging diffusion-based text-to-3D shape generation methods and convert it into a neural implicit representation, i.e., SDF, which is parameterized by neural networks. In the optimization stage, we design a data-driven differentiable

Figure 2: **Overall framework of our method.** Our method consists of two stages, initialization and optimization. We employ a 3D diffusion model to generate an initial mesh in the initialization stage. Then, the mesh is used to initialize the geometry network. Physics embedding is first computed to initialize the physics network in the optimization stage. Then, we sample 3D points to compute their geometry (SDF) and physics (displacement, strain, stress) properties. In this way, a total loss composed of geometry constraint loss, design loss, volume regularization loss, and physics loss is calculated to optimize the geometry and physics networks simultaneously.

physics layer to represent physics by neural networks implicitly. Then, the implicit physics representations are initialized by the FEM results computed for the initial 3D shape and our designed physics losses. On this basis, we further introduce relaxed geometry losses and an alternating training strategy to optimize the 3D geometry and its corresponding physics information collaboratively.

## 3.2 GEOMETRY REPRESENTATIONS AND INITIALIZATION

**Geometry Representations.** Our method consists of two key representations, geometry and physics. In order to allow our geometry to support physics perception, the geometry representation should have the following properties.

- *Differentiability of geometry*: The geometry representation should be differentiable to allow for end-to-end back-propagation during training.

- *Scalability of geometry*: The geometry representation should be scalable to allow for fine-grained sampling in the entire 3D space.

- *Discriminability of interior and exterior*: For physics, the solid mechanics will work on geometry surfaces and interior. Therefore, the geometry representation should make it easy to distinguish between interior and exterior.

To this end, we employ an implicit SDF Park et al. (2019) encoded by neural networks to represent our geometry. It is naturally differentiable and can be evaluated at different resolutions. For any 3D point $\mathbf{x}$, its SDF value $\hat{f}_S(\mathbf{x})$ represents the signed distance to the surfaces. That is, if the output SDF is a negative value, the corresponding point will be outside the object; otherwise, the corresponding point will be inside. Therefore, it is easy for point $\mathbf{x}$ to distinguish between interior and exterior with its SDF sign.

As physics calculations are contingent upon geometry, it's imperative to find a representation that the physics network can easily understand. Inspired by topology optimization Bendsoe & Sigmund (2013), we opt to convert the SDF to the density field function. This is achieved as $\hat{\rho}(\mathbf{x}) = \text{Sigmoid}(\frac{\hat{f}_S(\mathbf{x})}{\tau})$, where $\tau$ is the temperature hyper-parameter. Different from the SDF $\hat{f}_S(\mathbf{x})$, we follow the same assumption with "Solid Isotropic Material with Penalisation" Bendsøe (1989); Bendsøe & Sigmund (1999), i.e., the density field function $\hat{\rho}(\mathbf{x})$ will directly influence the physical properties of the material in the entire 3D space, such as Young's module and Possion's ratio Zienkiewicz & Taylor (2000). This definition allows incorporating the geometry information into the physics layer.

**Geometry Initialization.** To initialize our geometry representations, we use a 3D diffusion model to generate an initial 3D shape $\mathcal{M}$ based on the text prompts $\mathcal{T}$. In general, 3D diffusion models will output meshes to represent 3D shapes because mesh representations provide a clear topological structure, making them convenient for operations and analysis. To allow our method for the subsequent optimization with SDF, we convert the mesh $\mathcal{M}$ to an initial SDF $f_{init}$ by training a geometry network $G_\theta$ parameterized by $\theta$. This geometry network transfers the shape prior information from 3D diffusion models, thus we can focus on optimizing this geometry network in the next stage.

### 3.3 Physics Representations and Governing Equations

**Physics Quantities.** To describe the solid mechanics of geometry, we consider three fundamental physical quantities within the context of physics Bertram & Glüge (2015).

- *Displacement*: Representing the shift of any point within a geometric structure, displacement is characterized by a 3D vector $\mathbf{u}(\mathbf{x}) = (u_1(\mathbf{x}), u_2(\mathbf{x}), u_3(\mathbf{x}))$.

- *Strain*: Strain serves as a measure of deformation, depicting the relative displacement between points within a geometric body. Typically, strain at a point is denoted by a second-order tensor $\boldsymbol{\epsilon}(\mathbf{x}) = \{(\epsilon_{ij}(\mathbf{x})) | i, j = 1, 2, 3\}$.

- *Stress*: Stress encapsulates the internal forces exerted by neighboring points within a continuous geometric entity. The Cauchy stress tensor, often employed to describe stress, is denoted as $\boldsymbol{\sigma}(\mathbf{x}) = \{(\sigma_{ij}(\mathbf{x})) | i, j = 1, 2, 3\}$.

Moreover, it is important to acknowledge that the applicability of physical principles is often contingent upon specific external environmental factors, typically referred to as *boundary conditions*. In this study, the boundary conditions for a given geometry are defined based on typical usage scenarios associated with the object. For example, in the case of a table, an external force may be applied to its top surface in alignment with practical functional requirements.

**Governing Equations.** Governing equations serve as a foundational set of equations, often comprising ordinary or partial differential equations, to articulate the behavior of a physical system. As mentioned in Sec. 1, in this work, our focus lies primarily on the classical linear elastic system, which aptly characterizes elastic bodies under conditions of minimal deformation. The following equations delineate this system:

$$\boldsymbol{\nabla} \cdot \boldsymbol{\sigma} + \mathbf{F} = \mathbf{0}, \tag{1a}$$

$$\boldsymbol{\epsilon} = \frac{1}{2}[\boldsymbol{\nabla}\mathbf{u} + (\boldsymbol{\nabla}\mathbf{u})^\top], \tag{1b}$$

$$\boldsymbol{\sigma} = \mathbf{C} : \boldsymbol{\epsilon}. \tag{1c}$$

Here, $\mathbf{F}$ denotes body force per unit mass, ":" is the double dot product, and $\mathbf{C}$ is the fourth-order stiffness tensor. Note that, in this paper, we assume that a shape is made by the isotropic material. At this time, all elements in $\mathbf{C}$ are decided by only two parameters, Young's module and Poisson's ratio. Typically, these two parameters are determined by the material properties of the geometric body under consideration. The details can be found in Zienkiewicz & Taylor (2000). For the sake of simplicity and generality within this study, we set these parameters as 1 and 0.3, respectively. The ensuing boundary conditions supplement these governing equations:

$$\mathbf{u} = \bar{\mathbf{u}}, \quad \mathbf{x} \in \Gamma_u; \; \boldsymbol{\sigma} \cdot \mathbf{n} = \bar{\mathbf{F}}, \quad \mathbf{x} \in \Gamma_f. \tag{2}$$

Here, $\bar{\mathbf{u}}$ and $\bar{\mathbf{F}}$ represent the already known displacement and force, at the corresponding boundaries $\Gamma_u$ and $\Gamma_t$, while $\mathbf{n}$ denotes the unit outward normal vector on the relevant boundaries $\Gamma_f$.

**Physics Representations based on Neural Networks.** Similar to PINNs, in our physics layer, a neural network $U_\phi$ parameterized by $\phi$ is used to predict the displacement $\hat{\mathbf{u}}(\mathbf{x})$ at point $\mathbf{x}$ within the geometry. Subsequently, utilizing automatic differentiation based on Equation 1c, we obtain the predicted strain tensor $\hat{\boldsymbol{\epsilon}}(\mathbf{x})$. According to Equation 1a, we compute the predicted Cauchy stress tensor $\hat{\boldsymbol{\sigma}}(\mathbf{x})$.

### 3.4 PHYSICS PRIOR EMBEDDING AND TRAINING

**Physics Prior Embedding.** To enhance physics compliance guidance in geometry optimization, a high-precision simulator capable of providing accurate physical insights is essential. FEM stands out as a classical and reliable approach, renowned for its accuracy in capturing physical phenomena, particularly in simple systems such as linear elastic setups. However, FEM-based simulators often lack differentiability, which presents challenges when integrating them into online geometry optimization workflows. Physics-informed machine learning has recently emerged as a promising alternative to traditional physical simulations. The differentiability of neural network architectures enables seamless integration into online geometry optimization processes. Despite this advantage, physics-informed machine learning is still evolving, and current implementations face difficulties in achieving highly precise simulations for complex 3D geometries. To address this issue, we introduce a technique called physics prior embedding. This approach leverages the inherent differentiability of neural networks while enhancing the precision of physics predictions by incorporating FEM data into the training process of the physics-informed layers. By combining the differentiability of neural networks with the high-precision physical information provided by FEM, this technique facilitates effective geometric optimization, ensuring accurate physics guidance throughout the optimization process.

**Physics Driven Pretraining.** During the pretraining process, we keep the initial geometry unchanged and focus mainly on obtaining its precise physical information by utilizing a neural network. The whole process of the physics driven pretraining contains the following steps:

- *Data Generation*: We assume that a regular geometric space $\Omega^+$ (i.e., a cube) can adequately encompass the original geometry $\Omega$. Consequently, we sample a set of points $\mathcal{D}_{\Omega^+} = \{(\mathbf{x}^{(l)})\}_{l=1}^{N_{\Omega^+}}$ within $\Omega^+$ to evaluate whether the predictions provided by the physics layer satisfies the governing equations. Additionally, we require two other sets of sample points, denoted as $\mathcal{D}_{\Gamma_u} = \{(\mathbf{x}_{\Gamma_u}^{(l)})\}_{l=1}^{N_{\Gamma_u}}$ and $\mathcal{D}_{\Gamma_f^+} = \{(\mathbf{x}_{\Gamma_f}^{(l)})\}_{l=1}^{N_{\Gamma_f^+}}$. These sets contain sample points on the traction and traction-free boundaries of $\Omega^+$, respectively. Furthermore, we simulate the physics for the initial geometry using FEM, resulting in a precise-physics dataset $\mathcal{D}_{fem} = \{\mathbf{x}_{fem}, \mathbf{u}_{fem}, \boldsymbol{\sigma}_{fem}\} = \{(\mathbf{x}_{fem}^{(l)}, \mathbf{u}_{fem}^{(l)}, \boldsymbol{\sigma}_{fem}^{(l)})\}_{l=1}^{N_{fem}}$. This dataset serves to enhance the pretraining of the physics layer and provide better guidance for the optimization process.

- *Geometry Incorporation*: We integrate geometry information into the physics layer inspired by topology optimization techniques Yin et al. (2023). Specifically, it is achieved as follows:

$$\hat{\boldsymbol{\sigma}} = \hat{\rho}\mathbf{C} : \boldsymbol{\epsilon}, \tag{3}$$

and the equilibrium equation 1a is modified as:

$$\boldsymbol{\nabla} \cdot \hat{\boldsymbol{\sigma}} + \hat{\rho}\mathbf{F} = \mathbf{0}. \tag{4}$$

Thus, the predicted Cauchy stress tensor $\hat{\boldsymbol{\sigma}}$ can also be provided by combining the predictions from both the geometry network and the physics layer.

- *Pretraining*: To ensure that the physics layer provides highly accurate predictions for the physics, it must satisfy both the governing equations and the simulation results obtained from FEM. Therefore, the loss function for the pretraining process is defined as follows:

$$L_{pl} = w_{pde}L_{pde} + w_{bc}L_{bc^+} + w_{fem}L_{fem}, \tag{5}$$

$$L_{pde} = ||\boldsymbol{\nabla} \cdot \hat{\boldsymbol{\sigma}} + \hat{\rho}\mathbf{F}||_{\Omega^+}^2, \ L_{bc} = ||\hat{\mathbf{u}} - \bar{\mathbf{u}}||_{\Gamma_u}^2 + ||\hat{\boldsymbol{\sigma}} \cdot \mathbf{n} - \bar{\mathbf{F}}||_{\Gamma_f^+}^2,$$
$$L_{fem} = ||\hat{\mathbf{u}}_{fem} - \mathbf{u}_{fem}||^2 + ||\hat{\boldsymbol{\sigma}}_{fem} \cdot \mathbf{n} - \boldsymbol{\sigma}_{fem}||^2, \tag{6}$$

where $\Gamma_f^+$ contains all of the sample points on the traction force boundaries of $\Omega$ and the traction-free boundaries of $\Omega^+$, and $\hat{\mathbf{u}}_{fem}$ and $\hat{\boldsymbol{\sigma}}_{fem}$ are predictions provided by the physics layer for the sample points in $\mathcal{D}_{fem}$. In terms of the above loss functions, $L_{pde}$ is estimated based on $\mathcal{D}_{\Omega^+}$, $L_{bc}$ is estimated based on $\mathcal{D}_{\Gamma_u}$ and $\mathcal{D}_{\Gamma_f^+}$, and $L_{fem}$ is calculated based on $\mathcal{D}_{fem}$. These loss functions aim to make the predictions of the physics layer satisfy the PDE, the boundary conditions, and the precise physics data, respectively.

**Physics Driven Co-training.** We introduce a co-training framework for iterative geometry refinement to optimize the geometry while incorporating physical information. During this process, we aim to enhance the adherence of the geometry to objective physical laws while avoiding drastic changes that deviate from the intended concept and maintaining aesthetic qualities. Therefore, we introduce the following loss functions in our co-training process:

- *Design Loss*: Unlike conventional physics-informed machine learning, our objective extends beyond accurately computing the physics fields; we aim to achieve optimal geometry. Hence, we need to devise an indicator to assess the quality of a given geometry. The indicator we consider is based on achieving uniform stress distribution throughout the geometry, thereby minimizing stress concentration. Note that, though it is impossible to make the stress absolutely distributed in the whole geometry, such a way is still able to avoid the stress concentration as much as possible. Accordingly, the design loss is defined as $L_{design} = \max_{\mathbf{x} \in \Omega^+} \{\hat{\rho} \hat{\sigma}_{vm}\} - \frac{\int_{\mathbf{x} \in \Omega^+} \hat{\rho} \hat{\sigma}_{vm} \mathrm{d}\mathbf{x}}{\int_{\mathbf{x} \in \Omega^+} \hat{\rho} \mathrm{d}\mathbf{x}}$, where $\hat{\sigma}_{vm} = \sqrt{\frac{3}{2} \left( \hat{\boldsymbol{\sigma}} - \frac{\mathrm{tr}(\hat{\boldsymbol{\sigma}})}{3} \mathbf{I} \right) : \left( \hat{\boldsymbol{\sigma}} - \frac{\mathrm{tr}(\hat{\boldsymbol{\sigma}})}{3} \mathbf{I} \right)}$ is the approximated von-Mises stress, $\mathbf{I}$ is an identity matrix, and $L_{design}$ can be estimated based on $\mathcal{D}_{\Omega^+}$.

- *Geometry Constraint Loss*: The geometry constraint loss is designed to avoid excessive deformation of geometry. This is achieved by utilizing a dataset $\mathcal{D}_{gc} = \{(\mathbf{x}_{gc}^{(i)}, \rho_{gc}^{(i)})\}_{i=1}^{N_{gc}}$ sampled outside the envelope of the geometry. We assume that, for these points, the corresponding density values are kept unchanged. Thus, the geometry constraint loss can be calculated as $L_{gc} = \frac{1}{N_{gc}} \sum_{i=1}^{N_{gc}} \left( \hat{\rho}(\mathbf{x}_{gc}^{(i)}) - \rho_{gc}^{(i)} \right)^2$.

- *Volume Regularization Loss*: Similar to the geometry constraint loss, the volume regularization loss is to constrain the volume regularization of the optimized object to the target value. Its meaning is to ensure that the overall mass of the geometry is equal to or does not exceed a preset value. In this document, the volume regularization loss is defined as $L_{vr} = \max\{\int_{\mathbf{x} \in \Omega^+} \hat{\rho} \mathrm{d}\mathbf{x} - M_v, 0\}$, where $M_v$ is the maximum total volume. This can be estimated based on $\mathcal{D}_{\Omega^+}$.

- *Eikonal Loss*: To ensure the smoothness of the geometric surface, we also introduced the eikonal regularization, which can be described as $L_{eikonal} = (||\nabla \hat{f}_S|| - 1)^2$.

In addition to the aforementioned loss functions, $L_{pde}$, $L_{bc}$, and $L_{fem}$ are also incorporated into the co-training process. It's important to note that the dataset provided by FEM corresponds to the initial geometry. The associated physics values will also change as the geometry evolves during the co-training process. However, we assume that the changes in geometry are not substantial (we also employ $L_{gc}$ and $L_{vr}$ to achieve this), and the FEM data for the initial geometry can still offer guidance for optimization. Therefore, $L_{fem}$ remains employed throughout the co-training process.

The pseudo-code of the co-training process is shown in **Algorithm 1** of the **Appendix**. In this process, we alternately optimize two integrated loss functions $L_{pl}$ and $L_{cotrain}$. Specifically, $L_{pl}$ is the same as Equation 5, and $L_{cotrain}$ is calculated as follows:

$$L_{cotrain} = w_{design} L_{design} + w_{gc} L_{gc} + w_{vr} L_{vr} + w_{eikonal} L_{eikonal}, \tag{7}$$

where $w_{design}$, $w_{gc}$, $w_{vr}$, and $w_{eikonal}$ are the weights for different loss functions. Note that our focus lies on the convergence of $L_{pl}$ to ensure that the PINN can provide sufficiently accurate physical predictions, thereby ensuring that the physics can offer correct guidance for geometry optimization. Consequently, for every $t$ epoch, we optimize $L_{cotrain}$ for one step. Further details are outlined in **Algorithm 1** presented in **Appendix**, and $\alpha$ and $\beta$ are the learning rates in the optimization process.

## 3.5 DISCUSSIONS: DIFFERENCE FROM THE TRADITIONAL 3D GENERATION

In general, to verify the functionality of a generated 3D shape, its interaction with external environments must be considered. Physics dictates how a 3D object behaves in response to its surroundings. This is an aspect often overlooked in traditional 3D generation methods. Traditional 3D shape generation typically creates shapes based on training data, without accounting for the specific scenarios where the object will be applied or how it will interact with the physical environment. In contrast, Fun3D not only focuses on creating the object but also considers how it will interact with its physical surroundings and perform its intended function. This includes understanding how external forces

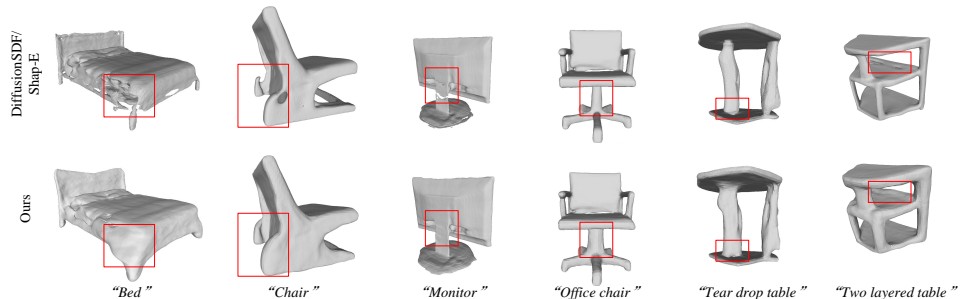

Figure 3: Qualitative geometry comparison.

act on the object, the resulting internal changes, and their overall impact on its performance. A noteworthy point in Fun3D is that the interaction between the object and its external environment must be well-defined. For example, objects like tables and chairs have straightforward interactions, with forces typically acting directly on their surfaces. In the following experimental discussion, we primarily verify the effectiveness of the proposed method on such 3D objects with clearly defined interactions. In contrast, for objects like animals, their interaction with the physical world is less explicit, making it harder to define how forces act on them. However, this does not imply that the proposed method cannot be applied to objects with less clear interactions. By operating under assumed interactive physical conditions rather than fully relying on the actual environment, the proposed method can still refine the generated 3D shapes. This capability is further validated in Sec. 4.5 using various animal objects.

## 4 EXPERIMENTS

### 4.1 EXPERIMENTAL SETTINGS

**Evaluation Metrics.** Our work points out that generated 3D shapes need to consider both geometric plausibility and functionality requirement. For the geometric plausibility, we use the Aesthetic Predictor LAION-AI (2022) to assess the aesthetic quality of 3D models following Huang et al. (2024). For the functionality requirement evaluation, we compute the maximal von-Mises stress of different 3D models to evaluate the durability of 3D models.

**Comparison Methods.** We compare our method with different text-to-3D generation methods, including Diffusion-SDF Li et al. (2023), Shap-E Jun & Nichol (2023) and Zero123 Liu et al. (2023a). Diffusion-SDF is trained on ShapeNet dataset Chang et al. (2015), Shap-E is trained on their own collected high-quality 3D assets, and Zero123 is trained on Objaverse dataset Deitke et al. (2023).

### 4.2 GEOMETRY COMPARISONS

We show the geometry comparison results with Diffusion-SDF and Shap-E in Figure 3. By giving a text description, Diffusion-SDF, Shap-E and our method will generate the corresponding object. We observe that Diffusion-SDF, Shap-E and our method can generate visually plausible 3D shapes. Moreover, our method can generate more complete and sturdy objects. For example, when generating a "two layered table", Diffusion-SDF generates a table with some holes. By incorporating physics knowledge into our method, our method needs to consider a more uniform stress distribution. Therefore, our method can generate a more complete 3D shape. When generating an object containing some complex connections, Diffusion-SDF and Shap-E produce fragile connections, like the connections in the red boxes of the "bed". In addition, the physics information helps our method generate more symmetric 3D shapes, like the "chair". Table 1 further reports the quantitative aesthetic evaluation of 3D models generated by our method and comparison methods. It shows that our method achieves better aesthetic scores, indicating that our generated 3D models are more visually appealing. Therefore, by incorporating precise physics into 3D generative methods, our method can generate higher-quality geometries, considering both visual realism and practical needs.

Table 1: The aesthetic quality (higher is better) and maximal von-Mises stress (lower is better) in the shapes generated by the Diffusion-SDF/Shap-E and our method.

| | Aesthetic ↑ | | Maximal von-Mises stress ↓ | |
|---|---|---|---|---|
| | Diffusion-SDF/Shap-E | Ours | Diffusion-SDF/Shap-E | Ours |
| Bed | 4.0387 | **4.2990** | 25.0583 | **5.9935** |
| Chair | 2.8905 | **3.1423** | 15.6467 | **5.8478** |
| Monitor | 3.1612 | **3.3115** | 215.9561 | **6.3204** |
| Office Chair | 3.7931 | **4.2783** | 40.5651 | **27.7762** |
| Tear Drop Table | 3.0500 | **3.6367** | 16.7696 | **8.0138** |
| Two Layer Table | 3.7178 | **3.7652** | 10.8793 | **2.7844** |

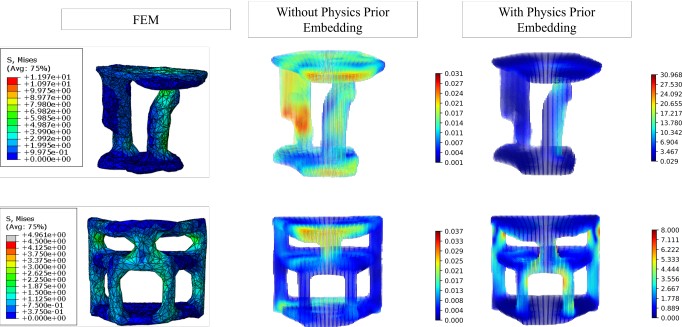

Figure 4: The physics predictions for the tear drop table (upper) and the two-layered table (lower) provided by the FEM (left), the physical layer without physics prior embedding, and the physical layer with physics prior embedding.

### 4.3 Analysis of Physics Conformity

To comprehensively evaluate the generated shapes from a physics perspective, we perform FEM simulations on shapes generated by both the comparison methods and our proposed approach. We obtain a stress field for each shape, from which we calculate the maximal von-Mises stress and record the results in Table 1. Analysis of the data in Table 1 reveals a consistent trend across all six shapes: the maximal von-Mises stress for shapes generated by our method is consistently lower than that of the shapes generated by the comparison methods. This indicates that, under identical loading conditions, our method yields shapes with reduced occurrences of stress concentration at specific points, mitigating the risk of localized high-stress levels. Consequently, the shapes generated by our method demonstrate enhanced load-bearing capacity and are less susceptible to damage. These findings underscore the efficacy of our proposed approach. Furthermore, in Figure 4 we present the predicted von-Mises stress provided by our physics layer for tear drop table and two-layered table, alongside the corresponding results obtained from FEM simulations. A comparison between the two reveals that, although the predicted von-Mises stress values are not identical to those obtained from FEM, they still exhibit stress distributions consistent with physical laws. In contrast, the physics layer fails to provide accurate and reasonable physical predictions without physics prior embedding.

### 4.4 Ablation Study

Here, we conduct additional experiments on the "curving chair" and "oval artistic table" to study the effect of different designs on our proposed method. The results are reported in Figure 5. We observe that the initial geometries generated by Diffusion-SDF Li et al. (2023) contain holes or weak joints, as shown in Figure 5(a). In contrast, by incorporating physics into our method, our generated shapes can significantly alleviate these defects. As for our different designs, without geometry constraints, the geometries cannot maintain the shape information. Without physics loss, the geometries become rough and uneven. Without FEM embedding, some details cannot be preserved. Without design loss, the geometries become fatter. These comparison experiments effectively demonstrate that our different designs facilitate our method to generate high-quality 3D models. Additionally, we quantitatively evaluate the results of the ablation study and assess the average aesthetic values and mean maximal von-Mises stress values in Table 2. In terms of aesthetic value, our method produces superior results, demonstrating its effectiveness in enhancing the visual appeal of the generated 3D shapes. Moreover, the full version of Fun3D improves the maximal von-Mises stress

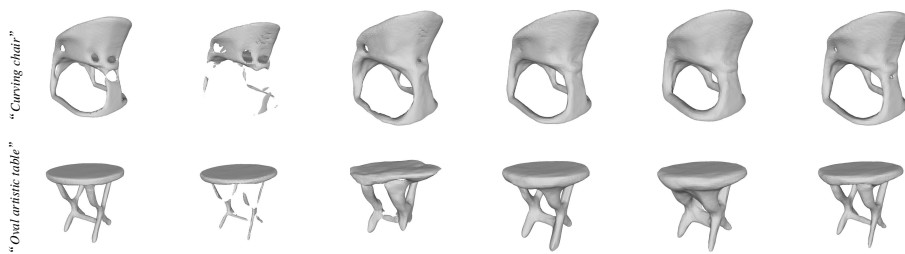

Figure 5: Additional ablation study results demonstrated that our proposed method eliminates the geometry defects introduced from Diffusion-SDF generation, strengthening the weak joints to make the optimized geometry more realistic and sturdy. The combined loss components work in harmony to ensure the shape integrity and sturdiness of the geometry. Hence, removing any lost component will result in degrading 3D shapes.

Table 2: Ablation study.

|  | Initial | w/o geometry | w/o physics | w/o FEM | w/o design | Ours (Full model) |
|---|---|---|---|---|---|---|
| Aesthetic ↑ | 3.7597 | 3.08775 | 3.2935 | 3.69965 | 3.48905 | 3.92305 |
| Maximal von-Mises stress ↓ | 26.22645 | - | 4.13275 | 4.89045 | 3.1852 | 7.3265 |

Figure 6: Qualitative comparison with Zero123 on animals.

compared to the initial shape. It is important to emphasize that maximal von-Mises stress assesses a 3D object's structural integrity under external forces from an engineering perspective, without considering artistic aspects. Consequently, while an algorithm without geometry loss, physics loss, and FEM embedding may yield better performance in terms of maximal von-Mises stress, the absence of these components can lead to a loss of aesthetic quality in the optimized shape.

## 4.5 GENERALIZATION AND DIVERSITY

In Sec. 4.2, we validate our method and comparison methods, Diffusion-SDF and Shap-E, on furniture, e.g., chair and table, which usually needs to consider both aesthetic and functionality in practice. To further investigate the efficacy of our method on generalization and diversity, we compare our method with Zero123 on animal objects. We relax the practical functionality requirement by simply applying force to the animal backs. In fact, animals do not always carry heavy objects on their backs. The results in Figure 6 show that our method can strengthen animal limbs, thus highlighting the functionality of our method in practice. This further demonstrates that our method can generalize to other text-to-3D generation methods and produce diverse 3D shapes, such as animals.

## 5 CONCLUSION

In this paper, we have proposed a physics compliance text-to-3D functional shape generation method to endow generated 3D shapes with physics perception capability. Our work reveals that the 3D shapes generated by existing text-to-3D generative methods do not conform to the laws of physics. Therefore, current generated 3D shapes are impractical in many real-world applications. To address this, we design a data-driven physics layer to learn physics information on demand when optimizing geometries. This guides our generated 3D shapes to satisfy both visual preferences and physical functionality, opening up a new avenue for text-to-3D shape generation.

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

# A APPENDIX

In this Appendix, we first provide the background knowledge of linear elastic solid mechanics in A.1. Then, the pseudo code is provided in A.2. In A.3, we deep dive into the mechanism of Fun3D, revealing its effectiveness in the loss function. A.4 details the implementation of Fun3D and A.5 studies the effectiveness of physical loss during the co-training progress. Then, in A.6, we present additional experimental analysis to verify generalization of our method. Next, wen report the computational time of our method in A.7. Finally, we discuss our method's limitation and future work in A.8.

## A.1 LINEAR ELASTIC SOLID MECHANICS

Though we introduced the governing functions of the linear elastic solid mechanics in Section 3.3, they may not be easy for the audience to understand without mechanical backgrounds. Henceforth, this section provides essential background knowledge of linear elastic solid mechanics.

Based on the assumption of isotropic homogenous material, both the Cauchy stress tensor and the strain tensor are symmetric, i.e., $\epsilon_{12} = \epsilon_{21}$, $\epsilon_{13} = \epsilon_{31}$, $\epsilon_{23} = \epsilon_{32}$, $\sigma_{12} = \sigma_{21}$, $\sigma_{13} = \sigma_{31}$, and $\sigma_{23} = \sigma_{32}$. In this way, the governing equations can be expanded as follows, which are called the strain-displacement equations, the constitutive equations, and the equilibrium equations:

- **Strain-displacement equations**: This set of equations establishes the relationship between the displacement field and the strain field, which are described as:

$$\epsilon_{11}(\mathbf{x}) = \frac{\partial u_1(\mathbf{x})}{\partial x_1}, \tag{8}$$

$$\epsilon_{22}(\mathbf{x}) = \frac{\partial u_2(\mathbf{x})}{\partial x_2}, \tag{9}$$

$$\epsilon_{33}(\mathbf{x}) = \frac{\partial u_3(\mathbf{x})}{\partial x_3}, \tag{10}$$

$$\epsilon_{12}(\mathbf{x}) = \frac{\partial u_1(\mathbf{x})}{\partial x_2} + \frac{\partial u_2(\mathbf{x})}{\partial x_1}, \tag{11}$$

$$\epsilon_{13}(\mathbf{x}) = \frac{\partial u_1(\mathbf{x})}{\partial x_3} + \frac{\partial u_3(\mathbf{x})}{\partial x_1}, \tag{12}$$

$$\epsilon_{23}(\mathbf{x}) = \frac{\partial u_2(\mathbf{x})}{\partial x_3} + \frac{\partial u_3(\mathbf{x})}{\partial x_2}, \tag{13}$$

- **Constitutive equations**: This set of equations mainly establishes the relationship between the strain field and the stress field:

$$\sigma_{11}(\mathbf{x}) = (\lambda + 2\mu) \cdot \epsilon_{11}(\mathbf{x}) + \lambda \cdot \epsilon_{22}(\mathbf{x}) + \lambda \cdot \epsilon_{33}(\mathbf{x}), \tag{14}$$

$$\sigma_{22}(\mathbf{x}) = (\lambda + 2\mu) \cdot \epsilon_{22}(\mathbf{x}) + \lambda \cdot \epsilon_{11}(\mathbf{x}) + \lambda \cdot \epsilon_{33}(\mathbf{x}), \tag{15}$$

$$\sigma_{33}(\mathbf{x}) = (\lambda + 2\mu) \cdot \epsilon_{33}(\mathbf{x}) + \lambda \cdot \epsilon_{22}(\mathbf{x}) + \lambda \cdot \epsilon_{33}(\mathbf{x}), \tag{16}$$

$$\sigma_{12}(\mathbf{x}) = \mu \cdot \epsilon_{12}(\mathbf{x}), \tag{17}$$

$$\sigma_{13}(\mathbf{x}) = \mu \cdot \epsilon_{13}(\mathbf{x}), \tag{18}$$

$$\sigma_{23}(\mathbf{x}) = \mu \cdot \epsilon_{23}(\mathbf{x}), \tag{19}$$

where $\lambda = \frac{E\nu}{(1+\nu)(1-2\nu)}$ and $\mu = \frac{E}{2(1+\nu)}$ are Lame parameters, and $E$ and $\nu$ are Young's module and Possion ratio, respectively.

- **Equilibrium equations**: This set of equations aims to describe the equilibrium relationship of forces, i.e.,

$$\frac{\partial \sigma_{11}(\mathbf{x})}{\partial x_1} + \frac{\partial \sigma_{xy}(\mathbf{x})}{\partial x_2} + \frac{\partial \sigma_{13}(\mathbf{x})}{\partial x_3} + F_1(\mathbf{x}) = 0, \tag{20}$$

$$\frac{\partial \sigma_{12}(\mathbf{x})}{\partial x_1} + \frac{\partial \sigma_{22}(\mathbf{x})}{\partial x_2} + \frac{\partial \sigma_{23}(\mathbf{x})}{\partial x_3} + F_2(\mathbf{x}) = 0, \tag{21}$$

$$\frac{\partial \sigma_{13}(\mathbf{x})}{\partial x_1} + \frac{\partial \sigma_{23}(\mathbf{x})}{\partial x_2} + \frac{\partial \sigma_{33}(\mathbf{x})}{\partial x_3} + F_3(\mathbf{x}) = 0, \tag{22}$$

where $\mathbf{F} = (F_1(\mathbf{x}), F_2(\mathbf{x}), F_3(\mathbf{x}))$ is the already known force field, e.g., gravitational field. For simplicy, we assume $\mathbf{F} = \mathbf{0}$ here.

Moreover, the boundary conditions can be described as follows:

- **Boundary Conditions**: The first is the surface stress boundary condition. It describes that the force should be at equilibrium at the boundary $\Gamma_f$:

$$\sigma_{11}(\mathbf{x}) \cdot n_1(\mathbf{x}) + \sigma_{12}(\mathbf{x}) \cdot n_2(\mathbf{x}) + \sigma_{13}(\mathbf{x}) \cdot n_3(\mathbf{x}) = F_{n_1}(\mathbf{x}), \tag{23}$$

$$\sigma_{12}(\mathbf{x}) \cdot n_1(\mathbf{x}) + \sigma_{22}(\mathbf{x}) \cdot n_2(\mathbf{x}) + \sigma_{23}(\mathbf{x}) \cdot n_3(\mathbf{x}) = F_{n_2}(\mathbf{x}), \tag{24}$$

$$\sigma_{13}(\mathbf{x}) \cdot n_1(\mathbf{x}) + \sigma_{23}(\mathbf{x}) \cdot n_2(\mathbf{x}) + \sigma_{33}(\mathbf{x}) \cdot n_3(\mathbf{x}) = F_{n_3}(\mathbf{x}), \tag{25}$$

where $\mathbf{n} = (n_1(\mathbf{x}), n_2(\mathbf{x}) n_3(\mathbf{x}))$ is the normal vector of a point $\mathbf{x}$ on the boundary $\Gamma_t$, $\bar{\mathbf{F}} = (F_{n_1}(\mathbf{x}), F_{n_2}(\mathbf{x}), F_{n_3}(\mathbf{x}))$ is the force at the boundary point $\mathbf{x} \in \Gamma_t$ (decomposed in the direction of the normal vector). The second one is called displacement boundary conditions, which means that the displacements of boundary points $\mathbf{x} \in \Gamma_u$ are already known, i.e.,

$$u_1(\mathbf{x}) = \bar{u}_1(\mathbf{x}), \tag{26}$$

$$u_2(\mathbf{x}) = \bar{u}_2(\mathbf{x}), \tag{27}$$

$$u_3(\mathbf{x}) = \bar{u}_3(\mathbf{x}), \tag{28}$$

where $\bar{\mathbf{u}} = (u_1(\mathbf{x}), u_2(\mathbf{x}), u_3(\mathbf{x}))$ is the already known displacement.

## A.2 PSEUDO CODE OF THE ALGORITHM

The pseudo code of our method is provided in **Algorithm 1**.

## A.3 THE MECHANISM OF FUN3D

In this section, we delve deeper into the effectiveness of different loss components and provide further insights into our implementation details. In Section 3.3, we introduced three groups of loss components: physics loss components ($L_{pde}$, $L_{bc}$, and $L_{fem}$), geometry loss components ($L_{gc}$, $L_{vr}$, and $L_{eikonal}$), and the design loss component $L_{design}$, which utilizes physical information to guide geometry optimization. In order to improve the sturdiness of the geometry and maintain its shape, we proposed the following optimization strategy. Initially, our implementation does not incorporate the design loss component. Instead, we rely on geometry loss components to facilitate the optimization, leading to an increase in the geometry volume. As the geometry reaches the target volume $M_v$, we limit the growth only to the high-stress regions based on the guidance of physics information. This is accomplished through a combination of $L_{design}$ and $L_{vr}$. Thus, in our implementation, we use the following simplified version:

$$L_{combine} = \delta \left( \max\{ \frac{\sum_{\mathbf{x} \in \mathcal{D}_{\Omega^+}} \hat{\rho} - M_v}{|\mathcal{D}_{\Omega^+}|}, 0\} \right) \cdot \left( \frac{\sum_{\mathbf{x} \in \mathcal{D}_{\Omega^+}} \{(\hat{\sigma}_{vm,max} - \hat{\sigma}_{vm})\hat{\rho}\}}{|\mathcal{D}_{\Omega^+}|} \right), \tag{29}$$

---

**Algorithm 1** Physics Driven Co-training

---

**Input:** $\mathcal{D}_{\Omega^+}, \mathcal{D}_{\Gamma_u}, \mathcal{D}_{\Gamma_f^+}, \mathcal{D}_{fem}, \mathcal{D}_{gc}, t$, and $epoch_{max}$;

**Output:** $G_\theta, U_\phi$;

 1: $epoch = 0$;
 2: **for** $epoch < epoch_{max}$ **do**
 3:    **if** $epoch \% t == 0$ **then**
 4:       Freeze the parameters of $U_\phi$;
 5:       Calculate $L_{cotrain}$ based on $\mathcal{D}_{\Omega^+}, \mathcal{D}_{\Gamma_u}, \mathcal{D}_{\Gamma_f^+}, \mathcal{D}_{fem}$, and $\mathcal{D}_{gc}$;
 6:       $\theta \leftarrow \theta - \alpha \cdot \nabla_\theta L_{cotrain}$;
 7:    **else**
 8:       Freeze the parameters of $G_\theta$;
 9:       Calculate $L_{pl}$ by utilizing $\mathcal{D}_{\Omega^+}, \mathcal{D}_{\Gamma_u}, \mathcal{D}_{\Gamma_f^+}$, and $\mathcal{D}_{fem}$;
10:       $\phi \leftarrow \phi - \beta \cdot \nabla_\phi L_{pl}$;
11:    **end if**
12: **end for**
13: Output the SDF function.

---

where $\delta(\cdot)$ represents the Dirichlet function, and $\hat{\sigma}_{vm,max} = \max_{\mathbf{x} \in \Omega^+} \{\hat{\sigma}_{vm}\}$ is detached during the training process. This combination serves a specific purpose: if the geometry surpasses the target volume, the design loss aids in limiting the growth or even removing the regions characterized by low von-Mises stress. Such regions are typically non-load-bearing and contribute minimally to the structure's functionality. This evolutionary approach offers several advantages. Firstly, it eliminates geometry defects inherited during the 3D model generation. Secondly, it fosters the development of a more sturdy geometry by aligning with principles of physics, thus enhancing the overall realism of the design.

### A.4 IMPLEMENTATION DETAILS

Our method is implemented by PyTorch Paszke et al. (2019) on two Teslia V100 GPUs with 32MiB graphics memory. The 3D diffusion model used in our method to obtain the initial 3D mesh is Diffusion-SDF Li et al. (2023). This model mainly generates two object types: chair and table. In addition, we also validate our method using other 3D diffusion model, Shap-E Jun & Nichol (2023) and Zero123 Liu et al. (2023a).

Our geometry network $G_\theta$ is modeled by an 8-layer multi-layer perceptron (MLP) with 256 hidden units. It is initialized by the geometric initialization Atzmon & Lipman (2020). To transfer the shape priors from 3D diffusion models to the geometry network, we follow DeepSDF Park et al. (2019) to train our geometry network. The geometry network is trained for 50 epochs, with each epoch comprising 200 training steps. A batch size of 16,384 SDF samples is employed, and the learning rate is set to 5e-4.

Our physics network $U_\phi$ is modeled by a 6-layer MLP with 125 hidden units. To train the physics networks efficiently and effectively, a dense-sparse strategy is employed to partition computation regions into "dense" regions where physics precision is crucial, and "sparse" regions where physics precision can be relaxed (i.e.regions distant from the mesh). The dense-sparse ratio is set at 0.3, with the dense regions being defined as regions where $f_{init}(\mathbf{x}) < 0.10$. The physics network is trained for 10,000 epochs with a learning rate 5e-3.

The co-training process spans 1000 epochs, with a learning rate of 1e-5 set for both the physics network and the geometry network. The weights corresponding to $L_{pde}, L_{bc}, L_{fem}, L_{combine}, L_{gc}$, and $L_{eikonal}$ are set to 1, 10, 10, 20, 10, and 10, respectively. The $t$ in Algorithm 1 is set to 10. In terms of the FEM data, we provide the displacement and stress values for the tetrahedral with 7000 nodes. All these data are obtained using Abaqus Barbero (2023).

Table 3: Comparison of Fun3D and Fun3D w/o physics update.

|  | Fun3D w/o physics update | Fun3D |
|---|---|---|
| Chair | 1.6456 | **1.4508** |
| Tear Drop Table | 13.5092 | **8.7364** |
| Two Layer Table | 2.5687 | **1.9049** |

### A.5 ADDITIONAL ABLATION STUDIES

In the previous ablation study, we verified the effectiveness of incorporating physics by comparing the proposed method with variants that omit physics loss or FEM loss. Here, we extend the comparison by introducing a stronger baseline that includes physical priors, focusing specifically on the effectiveness of the physical loss during the co-training process. The results are presented in Table 3. For the enhanced baseline, both FEM data and physics-informed loss were integrated into the physics driven pretraining process. However, in the physics driven co-training phase, the parameters of the physics-informed layer were fixed, and only the 3D shape was updated through the geometry network. We refer to this baseline as "Fun3D without physics update" and report the mean squared errors of the predicted stress fields for three optimized 3D shapes (the tear drop table, the two-layer table, and the chair). The error is calculated by comparing the physics-informed layer's predictions with the ground truth provided by the FEM method. The results show that for all three 3D shapes, the proposed full version Fun3D outperforms the baseline.

### A.6 MORE DISCUSSIONS ON GENERALIZABILITY

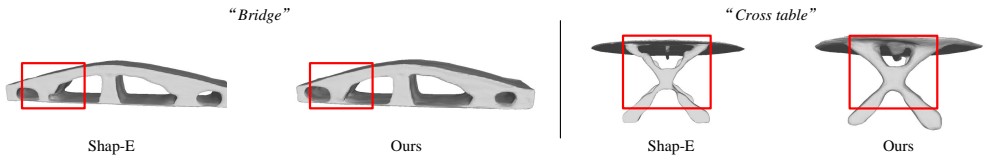

Figure 7: Our proposed method is not limited to only 3D shapes generated from Diffusion-SDF model, but also from the Shap-E model presented here.

To investigate the generalization of our method, we employ Shap-E as our 3D generative model to synthesize initial geometries. As presented in the results in Figure 7, when the generated 3D shapes have inherent defects, such as uneven surfaces or weak joints, our method can effectively eliminate these defects, resulting in more realistic 3D shapes.

### A.7 DISCUSSIONS ON THE COMPUTATIONAL TIME

Table 4: Time consumption of the proposed method.

|  | Bed | Chair | Monitor | Office Chair | Tear Drop Table | Two-Layer Table |
|---|---|---|---|---|---|---|
| Times | 411.6426 | 434.2527 | 436.2826 | 423.1809 | 422.5796 | 413.7805 |

One may be interested in the computational time required by the proposed method. Therefore, we discuss the time consumption involved in the shape optimization process in this subsection. On average, the joint optimization stage takes approximately 423.6 seconds to achieve optimization. For reference, the time spent on the joint optimization process for six different 3D shapes is listed in Table 4. In addition to the time required for joint optimization, precise physical data must be generated to initialize the physical network using the FEM method. Fortunately, for linear elasticity, FEM solvers are relatively efficient; in our experiments, the Abaqus solver took about 5 seconds to generate the necessary physical data. However, for more complex physical equations, such as hyperelasticity or dynamic physics, the FEM solver would require more time to complete the simulation.

## A.8 LIMITATION AND FUTURE WORK

While our method can generate high-quality 3D shapes with the assistance of precise physics data, there are still some limitations: 1) Efficiency. We use vanilla MLPs to parameterize our geometry and physics representations. This compromises the efficiency of our approach. Inspired by Instant-NGP Müller et al. (2022), we can leverage multi-resolution hash encoding to parameterize these two representations to improve our efficiency. 2) Precision of Physics. The precision of our physics information depends in part on the number of sampling points, following the similar reason explained in Chiu et al. (2022). In the future, we may increase the number of sampling points to boost the precision of physics information using more efficient geometry and physics representations.

