# OpenReview forum: "Looks Great, Functions Better: Physics Compliance Text-to-3D Shape Generation"
_ICLR.cc/2025/Conference — Submitted to ICLR 2025_

### Official Review · Reviewer_69Tk · 2024-10-17

**Soundness:** 2
**Presentation:** 2
**Contribution:** 2
**Rating:** 3
**Confidence:** 3

**Summary:**

The paper proposes a method to generate 3D shapes with increased physical plausibility. The pipeline is first initialized with the output of a traditional 3D generative model. The resulting shape is then optimized to have improved physical plausibility. In the examples, the method demonstrates how holes can be filled and weak support structures can made to be stronger.

**Strengths:**

I can't decide the strengths of the method based on the current submission. The results do not seem to be competitive and a better evaluation and presentation of the results would be necessary to judge the strengths of the method.

**Weaknesses:**

I would think the positioning and naming of the paper should be discussed. The paper tries to position itself as 3D generation paper, but to me it seems to be a post-process that is not specific to 3D generation and not integrated with the 3D generation. Therefore, it can be applied to any 3D model. As a result, the comparisons are just a demonstration that the post-process improves upon the initial model, but not a comparison against other work. The Design Loss, the Geometry Constraint Loss, and the Volume Regularization Loss do not directly, but only indirectly consider the generative model. While the paper tries to set itself apart from other competing work by claiming that “All these methods do not need to learn physics feedback during shape learning. Unlike them, our method collaboratively learns geometry and physics during training.”. I cannot see how this is true. Please confirm if you want to claim that you train a generative model that uses physics during training of the generative model. Physics and generation in your approach does not seem to be integrated in an interesting manner. This is still fine for a good paper, but the results would need to be a lot stronger. Overall, the quality of the results is not impressive and they do not seem to be useful in the current stage.

**Questions:**

I cannot find important information about the evaluation. Table 1 gives quantitative metrics, but how many shapes are used in the comparison and how are they generated? How did you ensure that an unbiased and fair set of shapes are used in the comparison? Cherry-picking would greatly influence the quantitative comparison, so it would be good to know how cherry-picking was avoided. This is a major issue.

I am not sure about the conversion of SDF to a density field. This will create a transition of densities from the outside to the inside, but it is not clear how fast this transition is compared to the scale of the object. Can you please describe the parameters of this conversion or let me know where I can find them in the paper. This is a minor issue.

**Details Of Ethics Concerns:**

N / A

---

> ### Author Response · Authors · 2024-11-18
> **Response to Reviewer 69Tk**
>
> *W. Positioning and naming of the paper.*
> A: Many 3D generation methods adopt a two-stage strategy to optimize 3D models, such as DreamGaussian and GaussianDreamer. Our method follows this two-stage pipeline, first generates coarse 3D mdodels using existing 3D generation models, then refines the 3D models with our geometry and physics co-training framework. For "All these methods do not need to learn physics feedback during shape learning. Unlike them, our method collaboratively learns geometry and physics during training.", we claim this in the Related Work Section and want to point out the previous methods (Mezghanni et al. (2021 and 2022); Liu et al. (2023b)) never use co-training framework to provide physics feedback for geometry learning.
>
> *Q1. Experimental details.*
> A: Six shapes are used in the comparison. The compared 3D models are generated by Diffusion-SDF or Shap-E. Based on these 3D models, we apply our physics-aware strategy to generate our 3D models. Different from traditional data-driven 3D generation methods, our method introduces interpretable solid mechanics to endow the generated 3D models with the ability to interact with the real environment (mainly reflected in the boundary conditions of the governing equation). In our experiments, we notice that unless the initial 3D models generated by our compared baseline methods are extremely unreliable (visually unappealing), our method can further boost the functionality of these 3D models. Based on our solid theoretical support and current reliable 3D generation baselines, we think that our comparisons in Table 1 are unbiased and fair.
>
> *Q2. Conversion of SDF to a density field.*
> A: In L211, we show that this conversion is achieved by $\hat{\rho}(\textbf{x}) = \text{Sigmoid}(\frac{\hat{f}_{S}(\textbf{x})}{\tau})$, where $\tau$ is the temperature hyper-parameter. As can be seen, the conversion is a sigmoid operation, which is very efficient.

---

> > ### Comment · Reviewer_69Tk · 2024-11-24
> > **Thanks for answering some questions**
> >
> > I think the concerns in my review still stand, and I will maintain my score.

---

### Official Review · Reviewer_abDh · 2024-11-03

**Soundness:** 2
**Presentation:** 3
**Contribution:** 3
**Rating:** 5
**Confidence:** 3

**Summary:**

The paper proposes Fun3D, a physics-informed text-to-3D generation method aimed at producing physically plausible 3D shapes based on text prompts. Existing text-to-3D models primarily focus on visual and geometric accuracy but lack physical realism, which limits practical applications. Fun3D addresses this gap by integrating physics, specifically solid mechanics, into the generative process. It uses a two-stage framework: an initial 3D shape is generated via 3D diffusion models and then optimized through a differentiable physics layer. This layer utilizes a mix of geometry and physics constraints, leveraging finite element method (FEM) data to improve stability and load-bearing capacity. Experiments demonstrate that Fun3D yields more physically robust shapes compared to baseline models like Diffusion-SDF, making it suitable for engineering and other real-world applications.

**Strengths:**

1. By incorporating solid mechanics and FEM-inspired optimization into 3D generation, the paper advances text-to-3D methods beyond visual realism, aiming for physical feasibility, which is valuable for applications requiring structural integrity.

2. The use of a neural network-based differentiable physics layer allows the system to be trained end-to-end, optimizing geometry while maintaining physics constraints.

3. The topic is important to real-world applications if we want to use a generative model to help produce solid objects.

**Weaknesses:**

1. Limited Comparisons: The authors only compare their method to Diffusion-SDF, neglecting recent advancements like SDFusion (CVPR 2023) and LucidDreamer (CVPR 2024), which utilize different geometry extraction and physics-informed components. Including these would provide a fuller assessment of the method’s capabilities.

2. Evaluation Metric Bias: Physical strength is evaluated using FEM, which is also an integral component of Fun3D’s training. This could bias results in favor of the proposed model. Additional evaluation metrics, such as load capacity or material distribution uniformity, could provide a more unbiased assessment of physical properties.

**Questions:**

1. Please address the concern about the weakness.
2. I am curious about the applicability of Fun3D to other types of objects where some of the parts are soft and some of the parts are solid. For example, in animals in Figure 6, the strength of the animal and the stress on them do not necessarily depend on the geometry we see on the outside. They are often more related to the structure of their bones and muscles. So why is Figure 6 shown or discussed in this paper, and why the proposed method can help animal generation to have better physical properties?
3. If some artist or architect wants to build something that is against the analysis of FEM, how do you balance the strength of the generated object and the design proposed by them?

---

> ### Comment · Reviewer_abDh · 2024-11-24
> **Official Comment by Reviewer abDh**
>
> The author only answered one of the four questions/weaknesses I posted in the general response and did not post any additional rebuttal or answer under my review comment. Also, I've read the general response and feel like the author did not want to discuss any of the issues brought up by other reviewers. Therefore, I am leaning more toward the reject side after the discussion period.

---

### Official Review · Reviewer_JvZN · 2024-11-03

**Soundness:** 3
**Presentation:** 3
**Contribution:** 2
**Rating:** 5
**Confidence:** 3

**Summary:**

The paper introduces a novel text-to-3d approach Fun3D that integrates the laws of physics into 3D shapes. By analyzing solid mechanics, it addresses the limitations of existing methods that produce impractical shapes. Fun3D utilizes 3D diffusion models and a differentiable physics layer to optimize shapes according to physical laws. The paper also demonstrates its high-quality generation results with physical functionality.

**Strengths:**

1. The paper introduces a novel perspective by integrating functionality with geometric and visual plausibility, significantly advancing the field of text-to-3D shape generation.
2. It achieves impressive results in generating 3D shapes, producing models that are both visually appealing and functionally viable compared with baseline methods.
3. The paper is well-written, presenting complex ideas with clarity and precision.

**Weaknesses:**

1. The initial mesh can be generated not only from a 3D diffusion model but also by state-of-the-art (SOTA) models based on multi-view diffusion model and reconstruction framework, such as InstantMesh. It is recommended to use these advanced mesh generation models in place of Shape-E for better performance and versatility.
2. The paper includes too few baselines in comparison, limiting the comparative evaluation. It would be beneficial to add more baseline models to provide a broader and more robust comparison of the proposed method's effectiveness.
3.There should be a broader range of methods to assess the physical laws in the generated meshes, rather than limiting the evaluation solely to compression tests.

**Questions:**

Please see above

---

> ### Comment · Reviewer_JvZN · 2024-11-24
>
> Thank you for the rebuttal. I keep my original score.

---

### Official Review · Reviewer_yJPi · 2024-11-03

**Soundness:** 2
**Presentation:** 3
**Contribution:** 2
**Rating:** 3
**Confidence:** 4

**Summary:**

The paper introduces Fun3D, a method for generating 3D shapes from text that ensures visual appeal and physical functionality. It features a data-driven differentiable physics layer for optimizing 3D shapes according to solid mechanics principles, providing the practicality of generated shapes for real-world applications.

**Strengths:**

Fun3D's primary advantage lies in its integration of physics-based constraints, which ensure that the generated shapes are not only geometrically sound but also practically functional in real-world scenarios.


The method employs a data-driven differentiable physics layer that allows for the simultaneous optimization of shape geometry and its physical attributes. This layer is trained using physics priors embedded through Finite Element Method (FEM) data, which provides a high level of accuracy in simulating the mechanical properties of the 3D shapes.

Fun3D adopts an alternating training strategy that efficiently balances the optimization of geometry and physics, leading to shapes that are robust and better suited for applications such as engineering design and virtual prototyping.

**Weaknesses:**

Primarily, the physical considerations in Fun3D are somewhat narrow, focusing mainly on stress, strain, etc, which is not a general framework and may not fully capture the complexity of real-world physical interactions. This could potentially lead to 3D models that, despite conforming to the considered physical laws, still lack a broader range of physical realism.

Additionally, the generated models' overall quality is reported to be average. The paper does not extensively discuss autoencoder-based 3D generation frameworks, which provide good geometric configurations and may already possess desirable physical properties. This omission raises questions about the comprehensiveness of the authors' exploration of the 3D generation landscape.

**Questions:**

I am interested in understanding how your Fun3D method compares to autoencoder-based 3D generation frameworks such as Craftsman and Clay, particularly in terms of physical characteristics. Do these frameworks already possess good physical properties, and if so, how does Fun3D's performance, especially in terms of physical functionality, qualitatively compare to these existing works?

I am curious about the practical validation of the physical properties of the 3D shapes generated by the Fun3D method. Could you elaborate on how you demonstrate that these shapes not only adhere to theoretical physical principles but also exhibit good physical characteristics in practical applications? Specifically, I would like to know if you have conducted any simulations or real-world tests, such as finite element analysis (FEA) simulations or actual 3D printing, to validate the structural integrity and functionality of the generated shapes in real-world scenarios.

---

> ### Author Response · Authors · 2024-11-18
> **Response to Reviewer yJPi**
>
> *W1, W2 and Q1. Physics generality and generalization to other 3D generation methods.*
> A: Thank you for your comments. Please see our overall response.
>
> *Q2. Simulation validation.*
> A: Thank you for your comments. In our paper, we have shown simulations using FEA, which we refer to as FEM. We present analysis of physics conformity in Section 4.3, which is based on our FEM simulation results shown in Table 1 and Figure 4. Our simulation results and analysis demonstrate the functionality of our generated shapes in real-world scenarios.

---

### Author Response · Authors · 2024-11-18
**Overall Response for All Reviewers**

We greatly appreciate the suggestions and comments provided by the reviewers for this paper. However, the reviewers seem to have misunderstood the core motivation of our work, particularly regarding the integration of physical information within the generative model. They appear to view the incorporation of physics as a trivial or inconsequential task. To address these concerns, we believe it is essential to provide a thorough response that clarifies the underlying motivations and the significant contributions of our approach.

Our primary objective is to explore the application of generative methods in more practical and engineering-oriented domains. While generative modeling has made remarkable strides, most advancements remain within computer vision, emphasizing visually appealing 3D models. However, in engineering, aesthetics alone are insufficient; models must meet rigorous safety and durability standards, which cannot be verified visually. This motivated our integration of physical information into the 3D generation process. Our approach, focused on solid mechanics, is **pioneering** and distinct from contemporaneous methods like PhysGaussian and Atlas-3D, offering a unique engineering perspective.

Additionally, we have observed that many reviewers are interested in validating the generalization ability of our method and comparing it with numerous generative models that do not incorporate physical considerations. However, we believe that such endless comparisons are not particularly meaningful, as our work primarily addresses two key points: 1) how to embed physics into the 3D generation process, and 2) the differences between the 3D models generated with and without the incorporation of physics, and the resulting impact of this integration. Regarding the first point, we have provided sufficient physics-related formulas to explain how to implement this process. For the second point, we have already demonstrated results and discussed the impact of incorporating physics in three models—Diffusion SDF, Shap-E, and Zero123. Therefore, we do not believe that continuing to add generative models and comparing them is a productive direction for our work. Moreover, based on our review of the original text recommended by the reviewer, it seems they expect us to further compare SOTA methods, which still fall under data-driven approaches and do not incorporate physical information embedding. The key distinction between physics-based generation and traditional data-driven generation lies in their interaction with the environment, as discussed in Section 3.5. When incorporating physical information, it is essential to consider the physical interactions between the 3D model and the real-world environment. A significant challenge posed by physical considerations is that, depending on varying physical conditions, even the same shape can lead to entirely different optimal results. While geometric data can be effectively captured in the dataset, external physical environments and their corresponding variations cannot be fully represented in the data. Consequently, even SOTA models that lack physical considerations may struggle to adapt to real-world physical environments.

Finally, we have observed that the reviewer suggests incorporating physics beyond solid mechanics to demonstrate the generality of our approach. However, this presents significant challenges. The physical models used in different fields, particularly those governed by differential equations, have been developed over decades or even centuries. As a result, each set of differential equations requires targeted consideration based on the specific characteristics of the field, making it difficult—if not impossible—to design a 'one-size-fits-all' approach that can handle all physics-related tasks. Furthermore, we do not believe such a universal approach is necessary. For any specialized engineering domain, embedding the relevant physical equations into the generative model is both meaningful and valuable, as it ensures that the model addresses the unique requirements and complexities of that field. It is also important to note that even current SOTA methods, such as PhysGaussian and Atlas-3D, primarily focus on a limited range of general elastic-plastic or hyperelastic models to produce visually appealing animations and striking effects, often overlooking engineering considerations. Nevertheless, although the aspect of generation was not fully addressed, we believe that both these SOTA physics-based approaches and our work contribute significantly to the advancement of future research in physics-aware generative modeling.

---

### Meta-Review · Area_Chair_6h7N · 2024-12-07

**Metareview:**

In this paper, the authors have proposed a physics-aware 3D shape generation method named Fun3D. The presentation of the paper is good, and the differentiable physics layer makes the pipeline end-to-end. However, this paper still have some significant limitations. As pointed out by the reviewers, the physical considerations are still narrow, the experiments are not sufficient enough, and the contributions of the paper should be better summarized. Due to the weaknesses above, the AC recommends a decision of rejection of the paper.

**Additional Comments On Reviewer Discussion:**

In the rebuttal, the authors highlight the differences with existing methods e.g. PhysGaussian and Atlas-3D, consider that it is not necessary to compare with more generative models, and introduce the challenges of the task. However, it does not address the concerns from the reviewers.

---

### Decision · Program_Chairs · 2025-01-22

Reject